# Sonodynamic Therapy for the Treatment of Intracranial Gliomas

**DOI:** 10.3390/jcm10051101

**Published:** 2021-03-06

**Authors:** Antonio D’Ammando, Luca Raspagliesi, Matteo Gionso, Andrea Franzini, Edoardo Porto, Francesco Di Meco, Giovanni Durando, Serena Pellegatta, Francesco Prada

**Affiliations:** 1Acoustic Neuroimaging and Therapy Laboratory Fondazione IRCCS Istituto Neurologico Carlo Besta, 20133 Milan, Italy; a.dammando89@gmail.com (A.D.); luca.raspagliesi@gmail.com (L.R.); matteo.gionso@st.hunimed.eu (M.G.); 2Neurosurgery Department, Fondazione IRCCS Istituto Neurologico Carlo Besta, 20133 Milan, Italy; edoardo.porto@gmail.com (E.P.); Francesco.DiMeco@istituto-besta.it (F.D.M.); 3Department of Health Sciences, University of Milan, 20122 Milan, Italy; 4Faculty of Medicine and Surgery, Humanitas University, Via Rita Levi Montalcini 4, 20090 Pieve Emanuele, Italy; 5Department of Neurosurgery, Humanitas Clinical and Research Center—IRCCS, 20089 Rozzano, Italy; andrea.franzini1@hotmail.it; 6Department of Neurological Surgery, Johns Hopkins Medical School, Baltimore, MD 21205, USA; 7Istituto Nazionale di Ricerca Metrologica I.N.Ri.M., 10135 Torino, Italy; g.durando@inrim.it; 8Laboratory of Immunotherapy of Brain Tumors, Unit of Molecular Neuro-Oncology, Fondazione IRCCS Istituto Neurologico Carlo Besta, 20133 Milan, Italy; serena.pellegatta@istituto-besta.it; 9Department of Neurological Surgery, University of Virginia Health System, Charlottesville, VA 22903, USA; 10Focused Ultrasound Foundation, Charlottesville, VA 22903, USA

**Keywords:** sonodynamic therapy, gliomas, 5-aminolevulinic acid, fluorescein, cavitation

## Abstract

High-grade gliomas are the most common and aggressive malignant primary brain tumors. Current therapeutic schemes include a combination of surgical resection, radiotherapy and chemotherapy; even if major advances have been achieved in Progression Free Survival and Overall Survival for patients harboring high-grade gliomas, prognosis still remains poor; hence, new therapeutic options for malignant gliomas are currently researched. Sonodynamic Therapy (SDT) has proven to be a promising treatment combining the effects of low-intensity ultrasound waves with various sound-sensitive compounds, whose activation leads to increased immunogenicity of tumor cells, increased apoptotic rates and decreased angiogenetic potential. In addition, this therapeutic technique only exerts its cytotoxic effects on tumor cells, while both ultrasound waves and sensitizing compound are non-toxic per se. This review summarizes the present knowledge regarding mechanisms of action of SDT and currently available sonosensitizers and focuses on the preclinical and clinical studies that have investigated its efficacy on malignant gliomas. To date, preclinical studies implying various sonosensitizers and different treatment protocols all seem to confirm the anti-tumoral properties of SDT, while first clinical trials will soon start recruiting patients. Accordingly, it is crucial to conduct further investigations regarding the clinical applications of SDT as a therapeutic option in the management of intracranial gliomas.

## 1. Conventional Therapies for Intracranial Gliomas

Intracranial gliomas are the most common primitive malignant neoplasms of the central nervous system, accounting for approximately 24.1% of all primary brain tumors [1].

Currently, therapeutic approaches mainly rely on surgical removal, which resolves local mass effects of the lesions and allows histopathological and molecular characterization; moreover, survival has been proven to increase with the extent of resection [2,3,4]. When surgery is not a viable option, characterization of the tumor may be achieved through a stereotactic biopsy procedure. Post-surgical management relies on conformational radiotherapy and pharmacological treatment, mainly with alkylating agents such as temozolomide (TMZ) and a procarbazine, lomustine and vincristine (PCV) combination scheme [3].

In recent decades major advances have been made relative to the survival of glioma patients, achieving longer Progression Free Survival (PFS) and Overall Survival (OS) [5]. However, the prognosis after a malignant glioma diagnosis remains extremely dire: glioblastoma, the most common and malignant subtype, harbors a three-years OS of 16% [5,6,7]. The outcome also depends on certain prognostic factors, encompassing histopathology, WHO grade, molecular features such as isocitrate dehydrogenase (IDH) and O6-methylguanine-DNA-methyltransferase (MGMT) promoter methylation, performance score and age at diagnosis [6].

Many factors are deemed responsible for such a dismal prognosis: on the one hand, surgery is generally unable to remove all the infiltrative foci of the disease, as not only are these not visualized by conventional imaging, but also extend up to several centimeters from the main tumor bulk within functioning brain tissue, so that the interest in preserving neurological function of the patients hampers the radicality of procedures [8,9]. On the other hand, the range of viable chemotherapeutic agents is quite restricted, as the blood-brain barrier (BBB) prevents most drugs from having significant effects in the central nervous system, especially at that infiltrative margin where the tumor’s mass effect does not disrupt the barrier [10,11]. Moreover, some authors suggest that tumor cells evading the resection of the main bulk are indeed Glioma Stem Cells, which are generally more resistant to post-surgical therapies and potentially involved in the tumor relapse [8].

The incurability of malignant gliomas is also influenced by the tumor microenvironment characterized by a high heterogeneity, a severe immunosuppression and subsequent low T cell infiltration [8]. 

Given the overall unsatisfactory yield of current treatment modalities, novel therapeutic approaches, including immunotherapeutic strategies, are continuously investigated to control tumor progression, eradicate infiltrative neoplastic cells and treat unresectable masses [3].

## 2. Sonodynamic Therapy: A New Asset for Non-Invasive Glioma Treatment

Ultrasound (US) is defined as mechanical waves with frequencies higher than audible sound (conventionally 20,000 Hz). Their characteristic ability to penetrate different types of matter without harm, including soft tissues, liquids and gases, make them particularly useful in diagnostic medicine; the echo-signals received back are recorded and displayed as grey-scale images, providing accurate anatomical information of the examined structures [12,13,14].

Therapeutic uses of ultrasound consist in inducing a wide range of biological effects by means of converging low-frequency US beams; though individual beams penetrate harmlessly through intervening tissue, their convergence to a confined small focus determines profound local effects, either thermal or non-thermal.

The latter effects, which predominate when low frequency, low power acoustic energy is administered at very short pulses (e.g., 5% or 10% duty cycle), are exploited by sonodynamic therapy (SDT) [13,15].

SDT employs the interaction between ultrasound and a non-toxic sono-sensitizing agent that selectively accumulates within the target tissue, offering the possibility of eradicating solid tumors in both a non-invasive and highly selective way; indeed, both the sensitizing agent and insonation are unable to produce relevant effects alone, which only occur when these elements are combined [16]. 

The mechanisms through which SDT exerts its cytotoxic effects on brain tumors are still widely unclear. Most accredited theories include cavitation effects, generation of Reactive Oxygen Species (ROS), induction of apoptosis in cancer cells, improvement of anti-tumor immunity, restraining of angiogenesis and induction of hyperthermia [17].

### 2.1. Cavitation Effect

The cavitation mechanism involves the formation and expansions of micro-bubbles filled with gas or liquid medium. Bubbles, in general, derive from gases dissolved in the medium or from pre-existing nuclei, like microbubbles injected as sonographic contrast agents. In detail, the negative “rarefactional” components of the ultrasound waves enable the expansion of small, stabilized gas-filled “cavities” or bubbles within a liquid medium [18].

Under the influence of ultrasonic pressure, these bubbles start oscillating, determining cell membrane vibrations or, if higher ultrasound intensities are applied, violent shock waves by collapsing, thus inducing mechanical lesions to surrounding tissue. This phenomenon contributes to the disintegration of water molecules and subsequent formation of hydroxyl radicals and hydrogen atoms [15,19,20,21].

### 2.2. Generation of Reactive Oxygen Species (ROS)

As firstly observed by Umemura, the scientific basis of SDT greatly relies on the generation of ROS through the simultaneous combination of low intensity ultrasound, molecular oxygen and a sensitizing drug. The concept is similar to the more established photodynamic therapy (PDT), in which light instead of ultrasound is used to activate the sensitizer [22,23].

Generation of ROS in SDT seems to be closely related to the cavitation effect: maximal expansion of gaseous bubbles and subsequent rapid implosion release considerable energy, resulting in raising temperatures and pressures in surrounding microenvironment; it has been suggested that these extreme temperatures and pressures at the point of implosion act as a “sono-chemical” reactor, which is able to generate ROS in the presence of water and oxygen; these unstable molecules can exert high cytotoxic effects if generated intracellularly, such as oxidative stress, DNA damage and apoptosis, and can induce lipid peroxidation if generated close to the cell membrane [19,24,25,26]. ROS production in this process is mainly explained through mechanisms of sonoluminescence and pyrolysis.

Sonoluminescence is a physical phenomenon that involves the emission of light from the implosion of bubbles suspended in a liquid medium when excited by ultrasound pressure [27,28].

The exact mechanisms by which light is generated after ultrasound irradiation of a solution is still uncertain. It may result from a combination of physical phenomena, such as the radiation of Bremsstrahlung, the argon rectification hypothesis and the recombination radiation. Bremsstrahlung radiation is an electromagnetic wave which is produced, according to the law of conservation of energy, when a charged moving particle, such as an electron, is decelerated by the deflection with another charged particle; the quantity of kinetic energy lost in this process is, thus, converted into radiation [29]. The argon rectification hypothesis postulates that high temperature inside the bubble determines the dissociation of molecular oxygen and nitrogen; the reactive species formed in the process are, then, irreversibly transferred to the liquid medium, leaving inside the bubble only non-reactive noble gases (such as argon, which is present in the air at a concentration of 1%); the latter are, then, responsible for bubble stability and electromagnetic emission [28,30,31]. Recombination radiation, finally, is the light radiation emitted when a free electron is captured by a ion [32].

Light from sonoluminescence may then activate sensitizers, determining production of ROS [23,33] in a way similar to PDT.

Pyrolysis theory postulates that localized increase of temperature during inertial cavitation process breaks apart the sonosensitizers, generating free radicals that can react with other endogenous substrates to generate ROS [23,34].

### 2.3. Induction of Apoptosis in Cancer Cells

Induction of apoptosis can be mediated by different mechanisms: down-regulation of Bcl-2 expression and up-regulation of BAX expression on the mitochondrial membrane; excessive production of ROS; calcium ions overload in the mitochondrial membrane and subsequent promotion of cytochrome-c release [35,36,37,38].

Under normal conditions, cells are able to clear a determined amount of ROS, as these are commonly produced in the organism during respiration; however, during SDT an excess of ROS is generated, which cannot be immediately cleared and induces oxidative stress inside the cells. Oxidative stress influences mitochondrial membrane potential, which may eventually lead to apoptosis. Moreover, SDT can alter the expression of two proteins located on the mitochondrial membrane and involved in cell apoptosis, with down-regulation of Bcl-2, whose expression protects against apoptosis, and up-regulation of BAX, whose expression promotes it. An increase in the synthesis of Bax protein results in the formation of BAX/BAX homodimers or BAX/Bcl-2 heterodimers by antagonizing Bcl-2. The resulting changes in proteins on the mitochondrial membrane increase the permeability of mitochondria and reduce its potential, inducing apoptosis [37,39].

Furthermore, SDT could promote calcium ion overload on the mitochondrial membrane and induce apoptosis of target cells by promoting cytochrome-c release and reducing the electric potential of the membrane itself [40,41].

### 2.4. Improvement of Anti-Tumor Immunity

There are different mechanisms through which SDT may enhance anti-tumor activity, promoting the transformation of macrophage inside the tumor from M2 to M1 type.

Microglia/macrophages are classified according to different functional states of polarization. M1 macrophages secrete pro-inflammatory cytokines and are involved in inflammatory processes, also playing a certain role in anti-tumor activity; M2 microglia/macrophages, on the other hand, reduce the inflammatory response, playing a major role in tissue repair. Malignant gliomas are strongly infiltrated by M2 microglia/macrophages involved in promoting an immunosuppressive microenvironment and supporting the tumor progression. Certain studies have reported that the latter macrophage population negatively correlates with prognosis in patients harboring a tumor. It was reported that mice treated with 5-aminolevulinic acid (5-ALA) SDT exhibit a greater amount of M1 CD68+ macrophage, compared with M2 CD163+ macrophage, whose number is significantly reduced [42,43,44,45].

Accelerating dendritic cells maturation in tumor microenvironment:

Wang et al. reported that expression levels of CD 68 and CD 80, markers for dendritic maturation on dendritic cells in vivo were significantly higher in mice treated with SDT than those in the control group, suggesting that this treatment may promote the maturation of dendritic cells, enhancing anti-tumor immunity [42].

In a melanoma murine study, a significant increase in IFN-g and TNF-a and decrease of IL-10 secretion were detected after SDT, contributing to anti-tumor activity [42]. Based on the importance of IFN-g and TNF-a in determining the M1 phenotype not only in macrophages but also in microglia, we can assume a potential role of SDT in a polarization from phenotype M2 to phenotype M1, inducing an anti-tumor and potentially phagocytic effector function [46].

In general, the effects exerted by SDT by targeting the tumor microenvironment and the tumor cells, on both infiltrating immune cells and tumor cells, can increase tumor immunogenicity. 

An immunogenic cell death induced in cancer cells can favor the emission of danger-associated molecular patterns (DAMPs), including Calreticulin (CRT), High Mobility Group Box 1 Release (HMGB1), Heat-Shock Proteins (HSP) 70 and 90, that can specifically recruit and trigger immune cells with effector function [47]. An alteration of cytokine/chemokine secretion pattern can direct T cell migration to the tumor areas along an inflamed gradient [48].

### 2.5. Restraining of Angiogenesis

Even if this mechanism is still not clear, anti-angiogenic effects of SDT with 5-ALA were observed by Gao et al. both in vitro and in vivo; they observed, in particular, that SDT using low-intensity ultrasound significantly inhibited proliferation and migration of endothelial cells in vitro, as well as the capacity of forming capillary networks; accordingly, in an in vivo study of human tongue cancer xenograft rodent model, they showed that the expression of vascular endothelial growth factor, a critical pro-angiogenic factor, was significantly reduced after treatment with SDT compared with subjects treated with ultrasound alone or controls [49].

### 2.6. Induction of Hyperthermia

Ultrasound-induced hyperthermia was shown to enhance the effect of SDT in some preliminary studies, although the exact mechanism by which this effect is exerted is still unclear [34,38,50]. Kinoshita et al. showed that intracellular protoporphyrin IX (PpIX) enhanced the cell-killing effect of hyperthermia, which can be produced by ultrasound exposure in a moderately acidic environment (pH 6.6) in vitro [34]. Kujawska et al. tested the thermal mechanism of SDT in vitro and demonstrated that it is crucial for viability of C6 rat glioma cells treated with ultrasound and 5-ALA [50].

## 3. Sonosensitizers for Brain Tumors

Sonosensitizers are chemical compounds whose particular therapeutic activity is triggered by ultrasound irradiation [51]. Particular interest in these chemotherapeutic agents as a potential approach for brain tumors has arisen in recent years, focusing on their safety and selectivity for tumor cells. At first, compounds known to have light-sensitive activity, referred to as photosensitizers, originally developed for PDT, were firstly considered due to their known non-toxic profile and selectivity for tumors. Early employment dates back to 1989, when it was recognized by the studies of Umemura and Yumita that hematoporphyrin, a well-known photosensitizer, could exert a cytotoxic effect also when irradiated by an acoustic field [22,52]. This evidence led to the assumption that ultrasound energy might determine, through a mechanism of sonoluminescence, electronic excitation in these light-sensitive compounds, transferring energy and starting a photochemical process which would eventually result in ROS generation. For these reasons, since Umemura’s preliminary experience, many other photosensitizers have been used to tackle brain tumors in combination with ultrasound [23,26]. 

Moreover [51,52,53,54], while light-based therapies are limited to superficial tumors, the greater tissue penetration of ultrasound waves through intervening tissues has opened a whole new field of application.

Overall, several sonosensitizers have been investigated in experimental contexts; in particular, two of these, 5-ALA and fluorescein, are already widely employed in current practice for guiding malignant brain tumor resection, due to their well-known selective accumulation in glial cells and good safety profile; these characteristics make them elective candidates for SDT experimental studies [55,56].

The main sonosensitizers studied for brain tumors are listed below and summarized in Table 1.

### 3.1. 5-Aminolevulinic Acid

5-ALA is currently the most employed porphyrin-based sonosensitizers for in vivo glioma models in SDT. 

5-ALA is a precursor of PpIX, an endogenous compound implied in heme synthesis which selectively accumulates in high-grade glioma cells due to reduced activity of ferro-chelatase [34,80,81]; the compound is therefore completely non-toxic per se but is able to exert cytotoxic and modulatory effects once “activated” by intersecting light or ultrasound waves. Indeed, other sensitizers have been observed to provide similar effects, among which are protoporphyrin IV, talaporfin and other compounds discussed below; however, the established use of this agent for fluorescence-guided surgery for high-grade gliomas (Figure 1) has placed 5-ALA in the spotlight for SDT [59].

First preclinical studies [57,58] regarding 5-ALA SDT safety demonstrated that no harm was exerted to surrounding healthy brain tissue using a peak power of 10 W/cm^2^ on a deep-seated glioma model; moreover, the treatment protocol highlighted a much greater reduction in tumor volume in the SDT group compared to controls. Bilmin et al. [60] reported that 5-ALA SDT caused marked cytotoxic effects in a rat RG2 glioma cell line, leading to decreased tumor cell viability, chromatin condensation and presence of apoptosis. 

The reduction of tumor cell viability was further confirmed by Ju et al. [38], who studied 5-ALA SDT in combination with hyperthermal-therapy at 42 °C; this combined approach was superior to both PDT and SDT alone in this study. 

The relationship between temperature and the outcome of SDT was further investigated by Wu et al., [61] who on the one hand confirmed the superiority of SDT in inhibiting C6 glioma cells’ growth in vivo compared to controls and sonicated-only subjects; a survival benefit for this group was also observed, while on the other hand higher baseline temperatures were not shown to significantly influence the outcome of the treatment.

In 2013, Yamaguchi et al. [63] proposed a new perspective on 5-ALA SDT: while most studies had until then employed frequencies around 1MHz, these authors were able to elicit tumor regression and inhibition of growth in an in vivo U87-MG glioma model with 5-ALA SDT using a significantly lower frequency of 25 kHz. Interestingly, this being the frequency used by most surgical ultrasonic aspirators, the authors advocated an intraoperative application of SDT, combining such instruments with 5-ALA. 

Conversely, Suehiro et al. [36] investigated the effects of 5-ALA-SDT at higher frequencies (2.2 MHz) in U87 and U251 glioma cells and in U251Oct-3/4 glioma stem-like cells, confirming the results previously reported. In vivo experiments, which only employed the U251Oct-3/4 line, showed that 5-ALA-SDT produced both necrosis in the focus area and apoptosis in the peri-focus area, decreasing the proliferative activity of the entire tumor without damaging the surrounding normal brain tissue. 

The glioma stem-like cell (GSC) model was used by Schimanski et al. in 2016 demonstrating for the first time the accumulation of PpIX after 5-ALA application. Tumor spheres growing in the absence of serum and in the presence of mitogenic growth factors (mostly EGF and bFGF) are considered as the appropriate way to grow cancer stem-like cells in vitro. GSCs can grow in vitro as neuro-spheres maintaining many molecular and genetic features of malignant gliomas from which they originate. In this context, the accumulation of PpIX after application of 5-ALA induced the acquisition of sensitivity to PDT [82]. 

A significant step forward has been made in 2019 by Yoshida et al. [62], who investigated the efficacy of 5-ALA SDT on malignant gliomas both in vitro and in vivo using the Exablate 4000 Insightec’s MRI-guided focused ultrasound (FUS) system (4000 J, 20 W for 240 s), which is currently the only device approved for high-intensity FUS ablation in the human brain. In addition, this study described an accumulation of 5-ALA-derived PpIX in the mitochondria of glioma cells after sonication, leading to the generation of ROS and, consequently, loss of mitochondrial membrane potential, cytochrome c release, caspase activation and apoptosis.

### 3.2. Fluorescein 

Fluorescein (2-(6-Hydroxy-3-oxo-(3H)-xanthen-9-yl) benzoic acid) is an organic compound belonging to xanthene-based dyes (Figure 1). Preferential accumulation in brain areas with impaired BBB and rapid washing out from vessels and healthy tissue are the main reasons for its selective accumulation in brain lesions, which makes this compound suitable for guiding malignant gliomas’ resection in neurosurgery [55,83].

When irradiated by ultrasound beams, this compound is transferred into an excited state, activating triplet oxygen and generating singlet oxygen, thus exerting its antitumoral activity. For such reason, as well as for its ability to selectively accumulate within tumor tissue, fluorescein-mediated SDT (FL-SDT) has been recently tested in vivo by Prada et al. on a rat-C6 glioma model [64]. This study found not only a selective accumulation of the compound in the subcutaneously injected tumor, peaking at 30′, but also a significant efficacy of FL-SDT compared to both untreated controls and subjects which received sonication without the sensitizer: indeed, the latter group only achieved a slower growth from the treatments, whereas FL-SDT subjects showed even a mild reduction in the tumor mass at the seventh day checkpoint. Surprisingly, in the same study no significant trends were observed relative to apoptosis markers or DNA fragmentation, although this was probably due to the late checkpoint [84,85].

### 3.3. Other Sonosensitizers

#### 3.3.1. Hematoporphyrin Monomethyl Ether (HMME)

Hematoporphyrin monomethyl ether (HMME) is a porphyrin-based compound, originally employed as photosensitizer; its high selectivity for tumor cells, among which gliomas, rapid removal and low toxicity make it suitable for SDT [41,70,86,87]. 

HMME was, indeed, the first drug to be investigated as a possible sono-sensitizing agent for SDT in 2008, when Li et al. observed in vitro significant growth inhibition, apoptosis rate and morphological changes in C6 glioma cells which had been incubated with HMME and subsequently sonicated, compared to the control groups. Later studies elaborated the same concept, also observing changes in the expression of several apoptosis markers, such as upregulation of Cytochrome C, Caspase 3, 9 and Bax, and downregulation of Bcl2, FasL. Overall, the occurrence of apoptosis was coupled to ROS production and mitochondrial membrane potential (MMP) loss, suggesting that the mitochondrial signal pathway may play a role in SDT-induced apoptosis in C6 glioma cells [68,70]. In 2011 Li et al. investigated the role of intracellular calcium release in HMME-SDT treated C6 rat glioma cells. An immediate and prolonged increase in intracellular Ca^++^ in the SDT group compared with control was described. Moreover, the addition of calcium chelators tempered all the results, strengthening the idea that calcium increase is a key element in the induction of SDT-mediated apoptosis [69]. These findings were confirmed by later studies that contributed to strengthen the correlation between calcium surge and apoptosis in C6 glioma cell lines [40,41] treated with HMME-SDT. 

HMME has also been implied as a sonosensitizer for the evaluation of sonodynamic therapy combined with photodynamic therapy (SPDT). This hybridized approach demonstrated that growth inhibition rate, apoptotic and necrotic markers and ROS production were highest in the SPDT group; the presence of ROS scavengers resulted in a decrease in the growth inhibition rate, underlining the pivotal role of ROS in both SDT and SPDT [67]. 

Only Song et al. have employed HMME-SDT in an in vivo glioma rat model. Tumor growth reduction was evaluated by MRI and was found highest in the SDT group. Histopathological alterations at 6–24 h after treatment highlighted lamellar necrosis, which was more pronounced in the SDT group compared to Ultrasound (US) alone. Immunohistochemical studies showed an increased expression of caspase 3 and cytochrome c, confirming the hypothesis that apoptosis is mediated by the mitochondrial pathway. However, the expression of MVD and VEGF protein were also much lower than in control groups, suggesting that the inhibition of angiogenesis and induction of ischemia, leading to the activation of the Bcl2 pathway, could also play a pivotal role in SDT-induced glioma cell apoptosis [66].

#### 3.3.2. Photofrin

Porfimer sodium is a hematoporphyrin derivative already widely used as photosensitizers in PDT for different types of cancer, due to its selective accumulation in tumor cells; ultrasonic irradiation triggers singlet oxygen production and apoptosis, which make it suitable for SDT in deep located lesions, such as brain glioma. Its main disadvantage is long-lasting persistence in skin, with resulting phototoxicity. 

Preliminary studies have revealed that Photofrin exerts good sonodynamic effects on glioma cells, and that ultrasonic irradiation with Photofrin generates ROS and triggers apoptosis in glioma stem-like cells. These are a rare subpopulation of self-renewing tumor cells with stem cell-like features, which aggressively migrate and may contribute to both malignancy and relapse after treatment. Photofrin-mediated SDT was effective in killing glioma stem-like cells, thus presenting itself as a promising treatment [75,76].

#### 3.3.3. Photolon

Photolon is the name of a chlorine e6 trisodium salt complex with polyvinyl pyrrolidone already in use for PDT. This compound selectively accumulates in cytoplasmatic organelles of tumor cells after administration and, despite being non-toxic in the absence of external activation, promptly shows marked cytotoxic activity through ROS production and cell death induction after ultrasound or light irradiation. Tserkovsky et al. reported the cytotoxic effects of Photolon-SDT on C6 glioma cells, as well as the possibility of enhancing the outcome of PDT through sonication of the same compound [77,78,79,88].

#### 3.3.4. Syno-porphyrin

Syno-porphyrin sodium (DVDMS) is a hematoporphyrin derivative based on Photofrin; compared to the latter, its high solubility in water determines short-term skin persistence. Different studies showed a greater sensitivity of DVDMS to both light and ultrasound irradiation, with excellent production of singlet oxygen, making it suitable for therapeutic application in different types of cancer, including glioma [89,90,91,92]. After administration, DVDMS preferentially localizes inside tumor cells mitochondria, probably due to misregulation of BBB patency. Indeed, DVDMS’ hydrophilic structure prevents it from crossing the intact BBB, thus most studies have employed different pharmacokinetic facilitators, such as liposomal carriers or BBB-disruptive sonication schemes [89,93,94].

Overall, four studies in vitro observed a selective accumulation of DVDMS compounds in glioma cells, which was generally superior in liposomal compounds than for the dye alone [71,72,73,74]. All studies reported higher rates of apoptosis, necrosis and growth inhibition in SDT schemes; sonication alone was inferior to sonodynamic therapy. In vivo models, employing either subcutaneous grafts [71,73,74] or intracerebral inoculation [72] of glioma cells, highlighted that SDT granted a significant survival benefit in treated groups compared to both controls and subjects who were only sonicated. Moreover, histopathological analyses observed elevation of apoptosis markers; in particular, Wen An et al. reported that both SDT and PDT reduced Bcl-x and p62 expression [74]. Interestingly, this study also found that fluorescence of the subcutaneous grafts decreased after both SDT and PDT, suggesting a photobleaching effect of both treatments, which would corroborate the sonoluminescence theory [74]. Only Pi et al. have employed a cerebral glioma model, which allowed them to observe far more significant survival increase and apoptosis markers in subjects also undergoing US-mediated disruption of the BBB [72].

#### 3.3.5. Bengal Rose

Bengal Rose (BR) is a xanthene-based dye whose cytotoxic effects can be triggered by ultrasound irradiation, determining ROS production; current fields of application include oncology and antimicrobial therapy.

Only one study has investigated the use of BR-SDT in an in vivo glioma model; this study reported tumor growth inhibition which was greater in the SDT group compared with the FUS only group. Surprisingly, safety evaluation found neither brain damage nor temperature rises in subjects insonated at 25 W/cm^2^, whereas this same US power was associated with greater thermal injury in most studies performed with different sensitizers. Furthermore, Yoshino et al. have reported that BR-SDT can successfully disrupt the BBB by breaking the tight junctions and increasing blood vessel permeability [65,95,96,97].

## 4. Final Perspective

High-grade gliomas are a group of primitive malignant tumors of the brain, whose aggressive behavior and high invasiveness determines frequent and early local recurrence, resulting in extremely poor outcome, notwithstanding optimal treatment with surgery, cytotoxic chemotherapy and radiotherapy [98,99].

Sonodynamic therapy has recently emerged as a potential new approach to high-grade glial neoplasia, both in addition to standard treatment for delaying tumor recurrences and as an alternative therapy for otherwise unresectable masses. Indeed, by exploiting the peculiar pharmacokinetics of sono-sensitizing agents, such as 5-ALA and Sodium Fluorescein, SDT offers the possibility of selectively narrowing cytotoxic and modulatory effects to glioma cells, while sparing surrounding parenchyma [16,23]; this is a crucial concept in neuro-oncology, since nervous tissue in proximity to neoplastic lesion might being involved in eloquent functions and should, therefore, absolutely be preserved in order to guarantee the patient’s quality of life.

Furthermore, SDT does not require physical access to the lesion nor to the brain, as surgical resection or PDT do, and can be repeatedly administered, which makes it suitable for achieving a longitudinal, long-term control of the disease; indeed, SDT has not been linked to ionization or stochastic effects compared to radiotherapy, as its mechanism of action rather involves peroxidation of membrane lipids via peroxyl radicals, physical destabilization of the cell membrane, and enhanced uptake of chemotherapy due to sonoporation [16]. Finally, unlike chemotherapeutic drugs, sono-sensitizing agents are per se completely non-toxic and pharmacologically safe.

Nevertheless, some major difficulties exist when delivering ultrasound through the intact skull; in fact, if ultrasound may be delivered without particular concerns through a craniotomy, the uneven bone thickness and composition throughout the skull may determine meaningful distortions in the ultrasound beams’ pathway, altering the target focus and hindering the effectiveness of the treatment; moreover, the high degree of ultrasound absorption can cause undesired bone heating, with the hazard of thermal damage to the surrounding brain tissue [100,101]. In order to overcome these issues, a large hemispherical phased array multielement transducer is employed to convey the beams, which is controlled by software that allows phase correction of single elements to restore the focal distortions caused by variations in skull thickness and density; moreover, low-frequency low-intensity ultrasound employed in SDT is generally transmitted without excessive energy loss due to transcranial absorption. Furthermore, scalp is refrigerated by continuously circulating water, which also acts as the coupling medium for ultrasound [102,103].

Notwithstanding these premises, current clinical studies are limited to other types of malignant tumors and still no study has assessed SDT impact on gliomas nor on any other type of intracranial tumor in human subjects, making this promising therapy still currently unavailable [104,105,106]. This is probably due to various sources of variability in treatment protocols, from dosage of the sensitizing agents to single features of sonication schemes, such as frequency, power and duration of exposure. 

To date, numerous preclinical experiments, both in vitro and in vivo, have been conducted which established the feasibility of inhibiting tumor growth and inducing apoptosis in glial tumors through SDT, as well as safety for adjacent tissues. The ability of SDT to induce immunogenic apoptosis, influencing anti-tumor activation, can be exploited to refine immunotherapeutic strategies.

Two particular compounds, already examined in different preclinical studies in glioma models for their known sono-sensitivity, are widely employed for guiding malignant brain tumors’ resection in neurosurgical practice: fluorescein and 5-ALA (Figure 1) [55,56]. Their known in vivo selective accumulation in glial cells and good safety profile make them elective candidates for further investigations and should be the first compounds to be focused on for clinical applications. The latter has gained sufficient experience to allow a first attempt at clinical translation of SDT. Indeed, the first-in-human clinical trial of MR-guided SDT with 5-ALA in glioma patients has recently been approved by the FDA and registered (NCT04559685), and one other is currently under review by the Italian Ministry of Health.

## Figures and Tables

**Figure 1 jcm-10-01101-f001:**
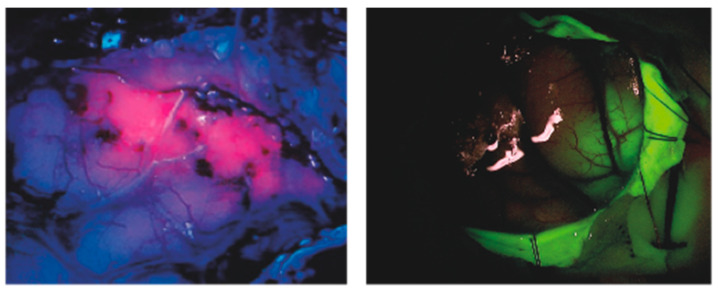
Intraoperative fluorescence images of 5-ALA (**left**) and Fluorescein (**right**) during glioblastoma resection. The known clinical safety, tumor-selectivity and sono-sensitivity of these compounds make them prominent candidates for brain Sono-dynamic Therapy (SDT).

**Table 1 jcm-10-01101-t001:** Summary of preclinical studies on Sonodynamic Therapy for gliomas.

Sonosensitizer	Title	Authors	Journal; Year
5-Aminolevulinic Acid	Sonodynamic therapy with 5-aminolevulinic acid and focused ultrasound for deep-seated intracranial glioma in rat	Ohmura et al. [57]	Anticancer Res. 2011 Jul;31(7):2527-33.
5-Aminolevulinic Acid	Sono-dynamically induced antitumor effects of 5-aminolevulinic acid and fractionated ultrasound irradiation in an orthotopic rat glioma model	Jeong et al. [58]	Ultrasound Med Biol. 2012 Dec;38(12):2143-50.
5-Aminolevulinic Acid	Porphyrin derivatives-mediated sonodynamic therapy for malignant gliomas in vitro	Endo et al. [59]	Ultrasound Med Biol. 2015 Sep;41(9):2458-65.
5-Aminolevulinic Acid	5-Aminolevulinic acid-mediated sono-sensitization of rat RG2 glioma cells in vitro	Bilmin et al. [60]	Folia Neuropathol. 2016;54(3):234-240.
5-Aminolevulinic Acid	Hyper-thermotherapy enhances antitumor effect of 5-aminolevulinic acid-mediated sonodynamic therapy with activation of caspase-dependent apoptotic pathway in human glioma	Ju et al. [38]	Tumour Biol. 2016 Aug;37(8):10415-26.
5-Aminolevulinic Acid	Enhancement of antitumor activity by using 5-ALA–mediated sonodynamic therapy (SDT) to induce apoptosis in malignant gliomas: significance of high-intensity focused ultrasound on 5-ALA-SDT in a mouse glioma model	Suehiro et al. [36]	J Neurosurg. 2018 Dec 1;129(6):1416-1428.
5-Aminolevulinic Acid	MR-guided Focused Ultrasound Facilitates sonodynamic therapy with 5-Aminolevulinic Acid in a Rat Glioma Model	Wu et al. [61]	Sci Rep. 2019 Jul 18;9(1):10465.
5-Aminolevulinic Acid	Sonodynamic therapy for malignant glioma using 220-khz transcranial magnetic resonance imaging-guided focused ultrasound and 5-aminolevulinic acid	Yoshida et al. [62]	Ultrasound Med Biol. 2019 Feb;45(2):526-538.
5-Aminolevulinic Acid	Low Frequency Ultrasonication Induced Antitumor Effect in 5-Aminolevulinic Acid Treated Malignant Glioma	Yamaguchi et al. [63]	J Cancer Ther. 2013;04(01):170-175.
Fluorescein	Fluorescein-mediated sonodynamic therapy in a rat glioma model	Prada et al. [64]	J Neurooncol. 2020 Jul;148(3):445-454.
Rose Bengal	Sonodynamic Therapy Consisting of Focused Ultrasound and a Photosensitizer Causes a Selective Antitumor Effect in a Rat Intracranial Glioma Model	Nonaka et al. [65]	Anticancer Res. 2009 Mar;29(3):943-50.
Hematoporphyrin Mono-Methil Ether	Study of the mechanism of sonodynamic therapy in a rat glioma model	Song et al. [66]	Onco Targets Ther. 2014 Sep 30;7:1801-10.
Hematoporphyrin Mono-Methil Ether	In vitro study of low intensity ultrasound combined with different doses of photodynamic therapy (PDT): Effects on C6 glioma cells	Li et al. [67]	Oncol Lett. 2013 Feb;5(2):702-706.
Hematoporphyrin Mono-Methil Ether	In vitro study of hematoporphyrin monomethyl ether-mediated sonodynamic effects on C6 glioma cells	Li et al. [68]	Neurol Sci. 2008 Sep;29(4):229-35.
Hematoporphyrin Mono-Methil Ether	In vitro stimulation of calcium overload and apoptosis by sonodynamic therapy combined with hematoporphyrin monomethyl ether in C6 glioma cells	Dai et al. [41]	Oncol Lett. 2014 Oct;8(4):1675-1681.
Hema-toporphyrin Mono-Methil Ether	Calcium overload induces C6 rat glioma cell apoptosis in sonodynamic therapy	Li et al. [69]	Int J Radiat Biol. 2011 Oct;87(10):1061-6.
Hematoporphyrin Mono-Methil Ether	Apoptotic effect of sonodynamic therapy mediated by hematoporphyrin monomethyl ether on C6 glioma cells in vitro	Dai et al. [70]	Acta Neurochir (Wien). 2009 Dec;151(12):1655-61.
Hematoporphyrin Mono-Methil Ether	Calcium Overload and in vitro Apoptosis of the C6 Glioma Cells Mediated by Sonodynamic Therapy (Hematoporphyrin monomethyl ether and ultrasound)	Hao et al. [40]	Cell Biochem Biophys. 2014 Nov;70(2):1445-52.
Sino-porphyrin	Theragnostic nano-sensitizers for highly efficient MR/fluorescence imaging-guided sonodynamic therapy of gliomas.	Liu et al. [71]	J Cell Mol Med. 2018 Nov;22(11):5394-5405
Sino-porphyrin	Sonodynamic Therapy on Intracranial Glioblastoma Xenografts Using Sino-porphyrin Sodium Delivered by Ultrasound with Microbubbles.	Pi et al. [72]	Ann Biomed Eng. 2019 Feb;47(2):549-562.
Sino-porphyrin	Tumor targeting DVDMS-nanoliposomes for an enhanced sonodynamic therapy of gliomas.	Sun et al. [73]	Biomater Sci. 2019 Feb 26;7(3):985-994.
Sino-porphyrin	Sino-porphyrin sodium is a promising sensitizer for photodynamic and sonodynamic therapy in glioma.	An et al. [74]	Oncol Rep. 2020 Oct;44(4):1596-1604.
Photofrin	Glioma stem-like cells are less susceptible than glioma cells to sonodynamic therapy with photofrin	Xu et al. [75]	Technol Cancer Res Treat. 2012;11(6):615-623.
Photofrin	The ABCG2 transporter is a key molecular determinant of the efficacy of sonodynamic therapy with photofrin in glioma stem-like cells. Ultrasonics	Xu et al. [76]	Ultrasonics. 2013;53(1):232-238.
Photolon	Effects of combined sonodynamic and photodynamic therapies with photolon on a glioma C6 tumor model	Tserkovsky et al. [77]	Exp Oncol. 2012;34(4):332-335.
Photolon	Photolon enhancement of ultrasound cytotoxicity	Tserkovsky et al. [78]	Exp Oncol. 2011 Jun;33(2):107-9.
Photolon	Imaging-guided focused ultrasound-induced thermal and sonodynamic effects of nano-sonosensitizers for synergistic enhancement of glioblastoma therapy	Wan et al. [79]	Biomater Sci. 2019 Jun 25;7(7):3007-3015.

## Data Availability

Data sharing not applicable. No new data were created or analyzed in this study. Data sharing is not applicable to this article.

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
