# Peer review of "Sonodynamic Therapy for the Treatment of Intracranial Gliomas"

_jcm, 2021, doi:10.3390/jcm10051101_

Round 1

Reviewer 1 Report

The authors review the use of sonodynamic therapy for the treatment of intracranial gliomas. The topic is interesting but the manuscript might be improved with attention to the following comments and questions:

It is curious that the agents named in table 1 for sonodynamic therapy are all known photosensitizers used for photodynamic therapy. This seems unlikely that they would be activated by both light and sound waves. This should be discussed.

The authors should discuss the many limitations of sonodynamic therapy for brain tumor therapy. The difficulty of human skull penetration, focusing the ultrasound,etc.

Author Response

We understand Reviewer #1 interest regarding the mentioned sensitizers and the fact that they were all formerly employed in photodynamic therapy; this concept was present in our paper, but not stressed much in our paper. Since photosensitizers have all a known safety profile, light-sensitive compounds were firstly investigated when SDT started to become an attractive modality for deep-seated intracranial tumors; the fact that the same compounds shared both light- and sound-sensitive activity was firstly reported in 1989 by the Japanese group of Yumita and Umemura (reference #52). Furthermore another reason for which photosensitizers were used relies on the fact that a possible mechanism for SDT is sonoluminescence, The creation  of light via ultrasound stimulation via different physical phenomena (radiation of Bremsstrahlung, argon rectification hypothesis, recombination radiation) may explain why photosensitizers are suitable for SDT as well. In our revised manuscript stressed this concept, which was present but probably unclear in the previous form. We also thank the Reviewer for letting us know that the limitations of SDT for the brain were not discussed in the previous manuscript. To this end we added a paragraph regarding limitations, and we hope to have addressed all the requested issues

Reviewer 2 Report

In this review, the authors have undertaken a large endeavour to describe and summarize currently available knowledge about novel and quite promising approach for treatment of malignant gliomas. They have compiled a very comprehensive review covering many aspects of sonodynamic therapy (SDT) – starting from potential mechanisms of action and types of sonosensitizers, to preclinical studies with the highlight of relevance of future clinical trials.

Could the SDT possibly be applied in the treatment of brain metastases caused by other types of tumors? Are there available any reports in the literature so far or any preclinical studies ongoing at the moment? If yes, please mention this future prospect of SDT in the context of important and challenging aspect of many human cancers in a few sentences within the section “Final perspective”.

There are some typos and misspellings in the manuscript.

Minor concerns:

Page 1, line 37 and 39 – an abbreviation SDT should be used instead of “sonodynamic therapy” in here

Page 1, line 37 – rather “clinical trials” than “clinical experiences”

Page 2, line 43 – no capital letter is needed in “Intracranial”

Page 3, line 117 – an abbreviation PDT should be explained when used for the first time and then used instead of “photodynamic therapy” within the text

Page 3, line 71 – what do the Authors have in their mind saying “tumor perpetuation”?

Page 2, line 81-84 – “Their characteristic ability to penetrate without harm different types of matter, including soft tissues, liquids and gases, made them suitable for several diagnostic applications in diagnostic medicine; the grey-scale map built from the echo-signals received back provides accurate anatomical information regarding the examined structures [12–14]” - to long and to complex sentence, should rather be divided into a few shorter sentences to make it easier to understand

Page 3, line 131-142 – to long sentence, should rather be divided into a few shorter sentences to make it easier to understand

Page 4, line 144 – an abbreviation of photodynamic therapy – PDT should be used in here

Page 4, line 168 and 172 – anti-tumor or antitumor? Please, use only one of them within the text

Page 4, line 175 and 180 – microenvironment or micro-environment? Please, use only one of them within the text

Page 4, line 178 – an abbreviation 5-ALA should be explained when used for the first time and then used instead of “5-aminolevulinic acid” within the text

Page 5, line 212 – an abbreviation ppIX should be explained when used for the first time in the text

Page 5, line 222 – an abbreviation PDT was used for the first time on page 3 so here should be used only PDT

Page 5, line 228 – an abbreviation 5-ALA should be used in here

Page 5, Table 1 – the title of columns in the table should be: Authors and Journal; Year

Page 8, line 236 – an abbreviation 5-ALA should be used in here

Page 8, line 249 – a caption of the Figure 1 – no capital letter is needed in “Glioblastoma”

Page 9, line 262-267 – to long and to complex sentence, should rather be rewritten and divided into a few shorter sentences to make it easier to understand

Page 10, line 327 – c6 should be corrected into C6 glioma cells

Page 10, line 344 – an abbreviation US should be explained when used for the first time

Page 10, line 350 and 356 – which of the forms: Photophrin or Photofrin is correct? Please, use only one of them within the text

Page 11, line 377 and 378 – an abbreviation BBB should be explained when used for the first time and then used instead of “blood brain barrier” within the text

Page 11, line 396 and 399 and 404 – BR or RB-STD? Please, use only one of them within the text

Page 11, line 401 – an abbreviation FUS should be explained when used for the first time

Page 11, line 405 – an abbreviation BBB should be used in here

Page 11-12, line 408 and 412 – High grade or high-grade? Please, use only one of them within the text

Page 12, line 442 – an abbreviation 5-ALA should be used in here

Author Response

We thank very much the Reviewer for this interesting comments. Manuscript had undergone a thorough linguistical revision and typos and misspellings were amended in the revised version of our manuscript. To the best of our knowledge there are still no clinical nor preclinical studies involving SDT for the treatment of intracranial metastases, as we just specified in the paper after your suggestion. Actually, in literature there are reports about clinical studies concerning extracranial tumors only, in particular regarding non-small cell lung cancer and breast cancer. For the aforementioned reasons, we strongly believe that Sonodynamic Therapy role in brain metastases should be investigated in the future at both pre-clinical and clinical level, but no reports exist in literature. We incorporated your suggestions in the revised version of the manuscript, and we feel that the paper appears now more exhaustive. 

Round 2

Reviewer 1 Report

My concerns have been addressed.